# The Effect of Socioeconomic Factors and Indoor Residual Spraying on Malaria in Mangaluru, India: A Case-Control Study

**DOI:** 10.3390/ijerph182211853

**Published:** 2021-11-12

**Authors:** Konrad Siegert, Welmoed van Loon, Prabhanjan P Gai, Jessica L Rohmann, Marco Piccininni, Anatol-Fiete Näher, Archith Boloor, Damodara Shenoy, Chakrapani Mahabala, Suyamindra S Kulkarni, Arun Kumar, Jacob Wedam, Pramod Gai, Rajeshwari Devi, Animesh Jain, Tobias Kurth, Frank P Mockenhaupt

**Affiliations:** 1Institute of Tropical Medicine & International Health, Charité–Universitätsmedizin Berlin, 13353 Berlin, Germany; konrad.siegert@googlemail.com (K.S.); welmoed.van-loon@charite.de (W.v.L.); prabhanjangai@hotmail.com (P.P.G.); fiete.naeher@web.de (A.-F.N.); jakob.wedam@charite.de (J.W.); 2SDM Research Institute for Biomedical Sciences, Shri Dharmasthala Manjunatheshwara University, Dharwad 580009, Karnataka, India; 3Institute of Public Health, Charité–Universitätsmedizin Berlin, 10117 Berlin, Germany; jessica.rohmann@charite.de (J.L.R.); marco.piccininni@charite.de (M.P.); tobias.kurth@charite.de (T.K.); 4Center for Stroke Research Berlin, Charité–Universitätsmedizin Berlin, 10117 Berlin, Germany; 5Robert Koch-Institute, 13353 Berlin, Germany; 6Kasturba Medical College, Mangalore, Manipal Academy of Higher Education, Manipal 576104, Karnataka, India; archith.boloor@manipal.edu (A.B.); drshenoy2001@hotmail.com (D.S.); chakrapani.m@manipal.edu (C.M.); animesh.jain@manipal.edu (A.J.); 7Karnataka Institute for DNA Research, Dharwad 580003, Karnataka, India; suyamindrask@gmail.com (S.S.K.); pramodbgai@gmail.com (P.G.); 8General Hospital Shikaripura, Shivamogga 577427, Karnataka, India; auaav4@gmail.com; 9District Hospital Mysore, Mysuru 570016, Karnataka, India; rajeshwaridevi14@gmail.com

**Keywords:** malaria, India, Mangaluru, urban, socioeconomic, migration, indoor residual spraying, directed acyclic graph

## Abstract

India faces 0.5 million malaria cases annually, including half of all *Plasmodium vivax* malaria cases worldwide. This case–control study assessed socioeconomic determinants of urban malaria in coastal Mangaluru, Karnataka, southwestern India. Between June and December 2015, we recruited 859 malaria patients presenting at the governmental Wenlock Hospital and 2190 asymptomatic community controls. We assessed clinical, parasitological, and socioeconomic data. Among patients, *p. vivax* mono-infection (70.1%) predominated. Most patients were male (93%), adult (median, 27 years), had no or low-level education (70.3%), and 57.1% were daily labourers or construction workers. In controls (59.3% male; median age, 32 years; no/low-level education, 54.5%; daily labourers/construction workers, 41.3%), 4.1% showed asymptomatic *Plasmodium* infection. The odds of malaria was reduced among those who had completed 10th school grade (aOR, 0.3; 95% CI, 0.26–0.42), lived in a building with a tiled roof (aOR, 0.71; 95% CI, 0.53–0.95), and reported recent indoor residual spraying (aOR, 0.02; 95% CI, 0.01–0.04). In contrast, migrant status was a risk factor for malaria (aOR, 2.43; 95% CI, 1.60–3.67). Malaria in Mangaluru is influenced by education, housing condition, and migration. Indoor residual spraying greatly contributes to reducing malaria in this community and should be promoted, especially among its marginalised members.

## 1. Introduction

India achieved major reductions in the burden of malaria in recent decades. Still, the country contributed to 3% of the global malaria cases, and 47% of *Plasmodium vivax* cases in 2018 [1]. In the last decade, India’s rapid economic and social transformation is accompanied by unique changes in the disease’s trajectory, such as the less pronounced decline or even increase in urban malaria [2]. In an effort to counteract the latter development, specific risk factors and policy measures have been applied, which require continuous re-evaluation [3].

While elsewhere, e.g., in sub-Saharan Africa, urbanisation generally was associated with lower malaria transmission [4], this is not necessarily universal. Within India, malaria is frequently imported from endemic to non-endemic states due to work-related migration. Poverty and lower education are established risk factors for malaria [5,6]. Labour migration of socioeconomically deprived individuals arises together with interlinked factors potentially affecting malaria risk, particularly in urban settings. Specific work and living conditions can increase risk. For example, urban construction sites close to stagnant water bodies can become mosquito breeding sites and expose workers on-site or in the nearby unprotected, provisional workers’ shelters. Housing conditions ranging from unsheltered night camps over temporary dwellings to regular accommodation differentially affect mosquito exposure [7]. Moreover, the presence and condition of living quarters overtly impact measures of vector management, i.e., bed nets, window nets, or residual insecticide spraying (IRS), one of the most effective control measures [8,9]. Incomplete coverage of health services for (migrant) construction workers and weak malaria surveillance may further aggravate their vulnerable situation [10,11]. Last but not least, it is well established that poverty and low levels of education negatively affect health-seeking behaviour, as well as access to treatment and prevention [12,13].

Mangaluru is a rapidly evolving harbour and business hub, located at the Arabian Sea in the southwestern Indian state of Karnataka. Between 1980 and 2010, the population in Mangaluru metropolitan area doubled [14]. In 2015–2017, Mangaluru contributed half of all malaria cases in the state of Karnataka, which itself accounted for 1% of the overall malaria burden in India [15]. In 2015, 10,920 malaria cases were recorded in Mangaluru [16]. In that year, we conducted a case–control study among malaria patients attending Mangaluru’s largest and governmental Wenlock District Hospital, many of whom were migrant workers [17] and community controls.

We aimed at estimating the effects of socioeconomic factors on malaria, in particular, education level, migrant status, housing conditions (with roof type as a proxy), and indoor residual spraying (IRS). These variables were chosen a priori by their presumed relevance to the local malaria acquisition pattern (labour migration) and for policymaking (education, housing, and IRS).

## 2. Materials and Methods

### 2.1. Study Design and Setting

The present unmatched case–control study with a group of community controls was conducted between June and December 2015, i.e., during peak malaria season, in Mangaluru, Karnataka, India. Mangaluru has approximately 485,000 inhabitants (agglomeration, 624,000) [14]. Malaria cases were recruited from the Wenlock District Hospital, a 1000-bed governmental hospital located in the city centre providing health care particularly for economically weaker members of the population. Recruitment details and clinical characteristics have been reported previously [17]. Briefly, symptomatic malaria patients (cases) were recruited at the outpatient malaria diagnostic unit upon microscopic diagnosis during the operating hours (08:00–16:00); those seeking care at other times were not included. Parasite density was assessed on Giemsa-stained thick blood films and counted on microscopy fields corresponding to 200 white blood cells (WBCs). Among patients, parasite density was calculated as parasites/µL using the parallel WBC count. Following DNA extraction (Qiamp blood mini kit, QIAGEN, Germany) from whole blood or dried blood spots (community controls), *Plasmodium* infection, and species were confirmed by nested polymerase chain reaction (PCR) assays [18].

Community controls were recruited between September and December 2015. The recruitment goal was 40 randomly selected healthy individuals in each of the 60 wards (census units) of the Mangaluru municipality. Community health workers of the Mangaluru City Cooperation visited randomly selected households in each ward during the daytime, and one volunteer per household was enrolled based on willingness to participate. Exclusion criteria were fever (axillary temperature ≥ 37.5 °C) and symptoms suspicious of malaria (e.g., headache, nausea). Finger-prick blood samples were collected from consenting control participants both on filter paper (Whatman 3MM Chromatography Paper, GE Healthcare Life Sciences, Freiburg, Germany) and on microscopy slides. Malaria diagnosis (in asymptomatic individuals) was performed identically to the one in cases (see above).

Regarding the sample size, recruiting 800 malaria patients and 2000 controls allows for detecting associations of rare exposures (10% prevalence) at strength as small as odds ratio (OR) 1.4, considering usual assumptions (80% power, 95% confidence interval). For exposures at 50% prevalence, this sample size facilities sufficient power to detect an OR as small as 1.2.

### 2.2. Questionnaire-Based Data Collection

Upon recruitment, all patients and control participants were interviewed by community health workers and completed a questionnaire on demographic and socioeconomic parameters including age, sex, education (none or below 10th grade, completed 10th grade, pre-university college, graduate, and above), religion, migrant status (i.e., region of origin, date of migration), occupation (e.g., construction worker, daily labourer (including coolie), housekeeping), number of persons in the household, number of rooms in the household, roof type (‘poor quality’ (i.e., hut, mud, straw, thatch, metal sheet), tiles, or cement), household possessions (i.e., electricity, fan, radio, television, fridge, motorcycle, bicycle), household income, preventive measures taken to avoid mosquito contact (mosquito bed net, mosquito repellent coil, skin cream, liquid, IRS within the past 6 months) and recent history of malaria (for participants and any household members).

### 2.3. Statistical Methods

Characteristics for included participants were summarised using median values and ranges, or frequencies and percentages, as appropriate. We identified relevant confounding variables that should be included in the models to obtain estimates of the total causal effects using directed acyclic graphs (DAGs) [19,20], built based on a priori knowledge (Figure A1). DAGs are increasingly implemented in applied epidemiology and promote the transparent presentation of underlying causal assumptions about the underlying data generation process [21]. Specifically, we were interested in quantifying the effects of four prespecified exposure variables. These included education level, being a migrant, IRS within the past 6 months, and housing conditions (operationalised as roof type). We, therefore, fit four separate logistic regression models with symptomatic malaria as the outcome and each exposure and corresponding set of confounders as covariates to control for the minimally sufficient confounding adjustment sets for each of these exposures. No interaction term was introduced in the regression models. The total effect of a given exposure variable (e.g., level of education) on the outcome (malaria) also includes indirect effects via mediating variables (e.g., income and occupation). According to the backdoor criterion [19,20], minimal adjustment sets determined from the DAG (Figure A1), which include the following:Education level on malaria (sex, age, religion);Migrant status (not born in Mangaluru) on malaria (sex, age, religion, education);Roof type on malaria (education, occupation, migrant status, household income);IRS within 6 months on malaria: (education, migrant status, roof type).

We further estimated the direct effects of education and migrant status on malaria in a secondary analysis. The direct effect is the effect of exposure of interest on the outcome excluding any indirect effects via mediators on the causal path from exposure to outcome [20]. In our study, the model to estimate the direct effect of education required adjustment for sex, age, religion, education, as well as household income, occupation, migrant status, roof type, malaria history, malaria history in the family, IRS in the past 6 months, mosquito net, window net, repellent coil, repellent liquid, and repellent skin cream. For the direct effect estimation of migrant status, the selected adjustment set was sex, age, religion, education, household income, occupation, education, roof type, number of people per room, malaria history, malaria history in the family, IRS in the past 6 months, mosquito net, window net, repellent coil, repellent liquid, and repellent skin cream. Effect estimates are reported as odds ratios (ORs) with corresponding 95% confidence intervals.

In an attempt to reduce possible bias introduced by missing data and to increase the power of our analysis, we imputed missing values using a multiple imputation approach [22], assuming an underlying missing value at a random mechanism [23]. In the multiple imputation process (10 datasets), we included the outcome variable (malaria), all exposure variables (education, roof type, migrant, IRS), confounding variables for all causal questions of interest (listed above), and variables affected by the monthly family income (e.g., availability of electricity in the house, possession of a fan, television, and fridge). For imputation, we used the originally collected, ungrouped categories for education, roof type, and occupation, the latter defined by the International Standard Classification of Occupations (ISCO). The analyses were conducted separately for each imputed dataset. In a second step, the final regression coefficients of interest were obtained by pooling the results across the 10 imputed datasets using Rubin’s rules [24].

As in all studies attempting to make inferences from observational data, interpretation of the regression coefficients as causal effect estimates relies on several important assumptions. In our case, relevant assumptions included positivity, consistency, conditional exchangeability, absence of measurement error, no model misspecification, and the rare outcome assumption [20].

We used R version 3.5.3 (R Foundation for Statistical Computing, Vienna, Austria) for all analyses, the MICE package [25] to generate the multiple-imputed datasets, and the DAGitty browser tool to produce the DAGs and to identify the minimal sufficient adjustment sets [26].

## 3. Results

Between June and December 2015, 909 malaria patients presenting at the Wenlock Hospital and 2478 community controls were enrolled in the study. Participants still in an educational track (i.e., under 18 years; *n* = 328 (40 patients, 288 controls)) and 10 homeless patients living on a boat were excluded from analysis (for consistency with control selection).

In total, 859 patients with microscopically visible and PCR confirmed *Plasmodium* infection, and 2190 community controls and were included in the present study analysis. Vivax malaria (70.1%) predominated over falciparum malaria (9.1%) and mixed-species infections (20.8%). The geometric mean parasite density was 3475/µL (95% CI, 3131–3858); 3.0% of patients were hospitalised. Among the controls, 90 (4.1%) PCR-confirmed asymptomatic *Plasmodium* infections were present, the uneven distribution of which is illustrated in Figure 1.

Basic demographic parameters and socioeconomic factors differed greatly between patients and controls (Table 1). Patients were younger than controls (medians, 27 vs. 32 years), and almost 95% of the patients were male compared with approximately 60% of the controls. The level of education was generally lower in patients than in controls. Almost 80% of the patients were not born in Mangaluru (migrants), in contrast to 35% of the controls. Among all migrants (aggregating patient and control data), 40% originated from elsewhere in Karnataka. Among those not originating from Karnataka, 70% originated from northeastern Indian states. The number of persons per household was higher among patients, with a range of up to 70 persons. More than half of the patients (56%) were either construction workers or daily labourers, compared with 41% among controls. The monthly family income (irrespective of household income) was lower in patients than in controls (medians, 6000 vs. 7000 Indian Rupee (INR)). Regarding housing conditions, most patients lived in a building with a cement roof (61%), followed by a tile roof (24%), and the fewest with poor-quality roofs (15%). Residing in a building with a tiled roof was most common among control participants (49%), followed by cement (41%), and poor-quality roofs (11%). IRS in the last 6 months was uncommon among patients (<3%), whereas more than half of the controls reported recent IRS. Wealth proxies such as access to electricity and possessions were all less common in the patient group. After imputation, the above-described characteristics and distributions remained similar (Table A1).

Results of the four models estimating the total effects of interest are displayed in Table 2. After adjustment for confounding, individuals who completed 10th grade had one-third of the odds of malaria, compared with individuals who had no education or an education level below 10th grade (aOR, 0.33; 95% CI, 0.26–0.42). For those completing higher levels of education (i.e., pre-university college, graduate education, or higher), the odds of malaria was halved (aOR, 0.50; 95% CI, 0.39–0.63). The direct effect analysis of education yielded similar results (no education or education below 10th-grade level, aOR, 0.45; 95% CI, 0.30–0.69; higher levels of education, 0.60, 0.37–0.97). Compared with residing in houses with a poor-quality roof, living under a cement roof conferred increased odds of malaria in the fully adjusted model (aOR, 1.99; 95% CI, 1.52–2.60), whereas a tiled roof slightly decreased the odds (aOR, 0.71; 95% CI, 0.53–0.95). We found a strong effect of reporting IRS within 6 months (aOR, 0.02; 95% CI, 0.02–0.04) compared with no reported recent IRS. Among migrants, fivefold higher odds of malaria were observed, compared with non-migrants (aOR, 5.0; 95% CI, 4.07–6.25, total effect). Examining the direct effect, after adjustment for the above-mentioned factors, the migration status conferred increased odds less pronounced (aOR, 2.43; 95% CI, 1.60–3.67) (Table 2).

## 4. Discussion

In this case–control study on urban malaria in Mangaluru, we observed considerable effects of education, IRS, roof type, and migrant status on the odds of malaria, after adjustment for confounding. Higher levels of education and, profoundly, IRS were protective as was, unexpectedly, poor-quality roofing (as compared with cement roofs). Migrant status increased the odds of malaria directly and indirectly via a set of mediating variables.

Completing 10th-grade education reduced the odds of malaria by more than two-thirds in the present study, both as total and direct effect, confirming respective findings on education and malaria reported globally [6]. Education is considered to act as a protective social determinant via symptom knowledge, increasing the likelihood of treatment-seeking behaviour and via multiple indirect pathways such as occupation and socioeconomic status [27]. The contribution of migration to malaria [28] is clearly discernible in our data from Mangaluru, with fivefold higher odds of malaria in the total effect model. Although observing a reduction in direct effects, migration can promote malaria itself, e.g., through imported infections or increased (genetic) susceptibility. However, our findings suggest mediating determinants, which, in turn, considerably increase malaria risks, such as income, occupation, housing, and crowded living condition, as well as poor coverage with IRS and other preventive measures. Another possible explanation of migration status being a risk factor for malaria is the potential influx of malaria susceptible migrants from low malaria transmission areas into a high malaria transmission area. However, in our study setting, this is likely not the case, because many migrants originated from areas where malaria prevalence exceeds that of Mangaluru [17].

Recent IRS substantially decreased the odds of malaria (>95%). Vector control is the main pillar in malaria control worldwide [29]. Our data indicate that more effective coverage of malaria control programs for migrant workers is required, as less than three percent of malaria patients reported IRS, while most patients were migrants working in construction sites. Many of them live in shelters with shared sleeping spaces and, thus, have increased mosquito exposure [30]. Malaria control programs in India often aim at the residential population, possibly neglecting migrant workers as uncontrolled reservoirs of urban transmission [31] including drug-resistant *Plasmodium* strains [32]. Our data thus highlight the importance of vector control to be expanded to the poor and hard-to-reach populations, particularly, migrant workers.

In a context of very diverse housing conditions ranging from rudimental shelters to urban condominiums, we chose the roof type as an overall proxy parameter for the quality of housing. While residing in living quarters with a tiled roof decreased the odds of malaria, surprisingly, a cement roof doubled the odds of malaria compared to living in residences with poorer quality roof types (i.e., hut, mud, straw, thatch, or metal sheets). Prior studies suggest that—in rural areas—traditional housing bears an increased risk for malaria [33], and another meta-analysis reported that modern roofs are malaria protective in both rural and urban settings [34]. A possible explanation for our counterintuitive finding is that concrete roofs only contribute to reduced mosquito entry when other preventive measures such as eave blockings or window screens are in place [35]. Additionally, cement roofs could, by themselves, provide stagnant water bodies serving as mosquito breeding sites [36]. We did not collect the necessary more detailed information to address these possible aspects, which may result in some residual confounding.

In our analysis, we applied a causal framework using an a priori defined causal structure [19,20,37]. The structure and implied assumptions are described in the DAG in Figure A1 and justified in Table A1. Having no connection between two variables in a causal DAG is a sharp null hypothesis of no causal effect on any of the individuals in the population [20]. Consequently, whenever in doubt, we conservatively included potential causal relationships, even if the evidence is not fully certain. For example, we assumed that relations between socioeconomic determinants affect malaria (e.g., religion -> malaria, via mosquito bite exposure depending on clothing or behaviour after sunset). An additional possibly relevant variable, early life socioeconomic status [27], could have contributed to unmeasured confounding, but since it was not measured in this study, we could not include it in our models. However, we believe that education and malaria history in combination with the other, included, socioeconomic parameters may serve as a proxy for this variable.

Another limitation is a potential selection bias of control participants since they were recruited during the daytime hours at home (e.g., males tended to be away for work), while patients presented to the hospital were more frequently males and construction workers or daily labourers. Moreover, the community controls were randomly recruited among households in the city of Mangaluru, whereas recruited patients came also from nearby rural areas and did not always live in a classic household. We attempted to address this problem by excluding patients without a residence (homeless) from the study since this heterogeneity was not captured among included control participants. A deficit in our data is the lack of IRS coverage by neighbourhood, which could have provided a better insight into the effect of IRS since such vector control can have an effect beyond individual households. We further acknowledge that the reliability of the data on self-reported socioeconomic determinants obtained via interviews may have led to potential missing not-at-random mechanisms. Furthermore, assessing socioeconomic status by scale or questionnaire has inherent limitations [6].

## 5. Conclusions

In conclusion, being a migrant in Mangaluru, coastal southwestern India, contributes to the urban malaria incidence directly and indirectly via mediating factors, such as housing conditions and occupation. Education level, roof type, and recent IRS have a substantial, mitigating effect on malaria. Our results suggest that urban malaria incidence in this setting is, at least in part, driven by the poor living conditions of migrant workers. Improved IRS coverage of the poor and marginalised would likely reduce the burden of malaria in this urban community. This appears both feasible and necessary considering the operational possibilities of an emerging Indian business hub, on the one hand, and the malaria-related losses, on the other.

## Figures and Tables

**Figure 1 ijerph-18-11853-f001:**
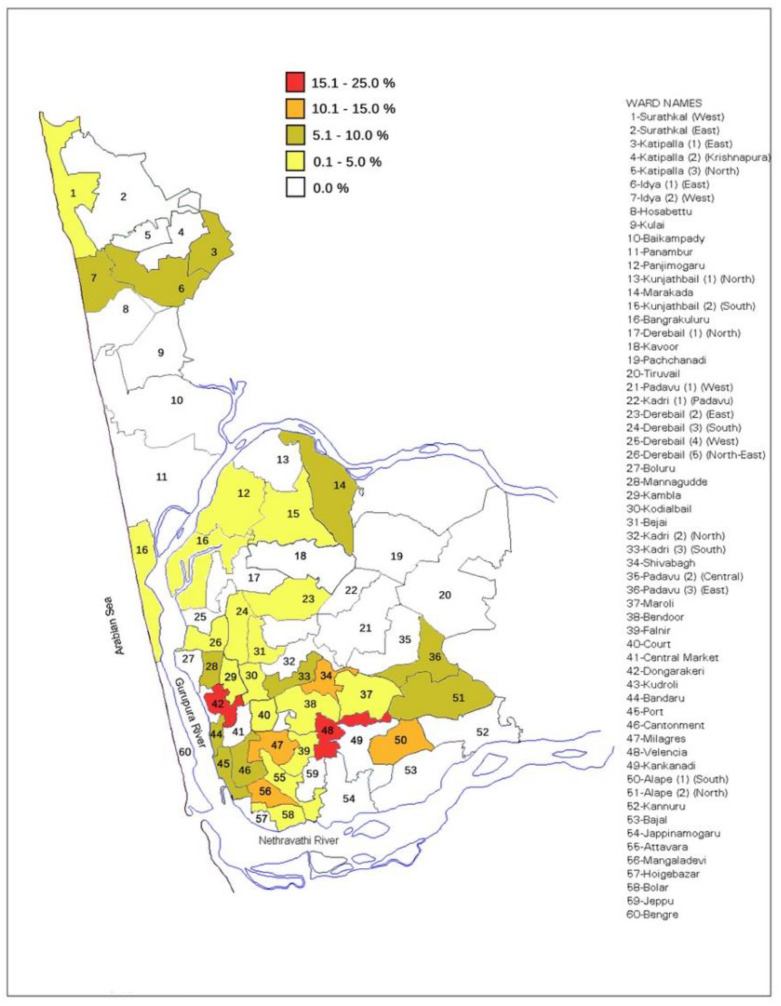
*Plasmodium* prevalence among randomly selected, asymptomatic controls in Mangaluru, 2015, by wards (census units).

**Table 1 ijerph-18-11853-t001:** Descriptive statistics of the study population, including demographic and socioeconomic variables ^1^ used for analysis and/or imputation.

	Controls (N = 2190)		Patients (N = 859)	
	% or Median (Range) and Mean (SD)	n/N or N	% or Median (Range) and Mean (SD)	n/N or N
Male	59.3	1297/2186	93.6	804/859
Age, years	32 (18–94); 35.2 (13.7)	2161	27 (18–82); 31.0 (11.4)	859
Parasite species				
*p. vivax*	3.6	78/2190	70.1	602/859
*p. vivax/falciparum*	0.2	5/2190	20.8	179/859
*p. falciparum*	0.3	7/2190	9.1	78/859
None	95.9	2100/2190	0.0	0/859
Religion				
Hindu	80.0	1699/2125	74.5	639/858
Muslim	15.2	324/2125	20.9	179/858
Christian	4.0	86/2125	4.2	36/858
Other	0.8	16/2125	0.5	4/858
Education level				
None or below 10th grade	54.5	1121/2056	70.3	596/848
10th grade	26.1	536/2056	14.4	122/848
Pre-university college, graduate or above	19.4	399/2056	15.3	130/848
Household income, INR	7000 (2–100,000); 7133 (5294)	1425	6000 (0–35,000); 7664 (4194)	841
Construction worker or daily labourer	41.3	867/2099	57.1	487/853
Migrant	35.3	714/2024	78.1	669/857
Roof type				
Poor quality	10.6	220/2085	15.1	128/845
Tiles	48.8	1018/2085	23.7	200/845
Cement	40.6	847/2085	61.2	517/845
Number of persons in household	4.0 (1.0–18.0); 4.8 (2.0)	2081	5.0 (1.0–70.0); 6.9 (6.5)	844
Number of rooms in household	3.0 (1.0–13.0); 3.3 (1.5)	2011	1.0 (1.0–11.0); 1.6 (1.2)	826
Number of persons per room in household	1.5 (0.1–10.0); 1.7 (1.1)	2005	4.0 (0.2–70.0); 5.5 (5.8)	824
Electricity	98.5	2003/2033	95.7	816/853
Fan	94.0	1827/1943	71.2	607/853
Television	87.2	1647/1888	19.5	166/853
Refrigerator	65.4	992/1516	5.2	44/853
Motorcycle	33.8	377/1117	3.5	30/853
Radio	36.7	440/1200	2.6	22/853
Bicycle	21.6	230/1064	1.6	14/853
Ever had malaria before	15.9	318/2000	46.9	400/853
Had household member with malaria	12.5	257/2052	25.3	216/853
Stagnant water bodies near the house	4.5	89/1986	31.0	264/852
Slept under mosquito net last night	57.5	1177/2048	38.9	332/853
Window net	44.7	906/2028	3.9	33/853
Use of mosquito repellent liquid	74.8	1194/1597	9.7	83/853
Use of mosquito repellent skin cream	17.1	163/954	1.3	11/853
Use of mosquito repellent coil	66.3	1028/1551	38.8	331/853
Indoor residual spraying last 6 months	52.7	780/1480	2.6	22/853

^1^ For all variables, *p* ≤ 0.001 using a two-tailed Fisher’s test or a Mann–Whitney test as applicable; INR, Indian Rupee.

**Table 2 ijerph-18-11853-t002:** Effect estimates for malaria in Mangaluru, India.

Variable of Interest	OR	CI (95%)	Adjusted for Variable Set	aOR	CI (95%)
Level of Completed Education			Sex, Age, Religion		
None or below 10th grade	1	-ref-		1	-ref-
10th grade	0.43	0.34–0.54		0.33	0.26–0.42
Pre-university college or graduate and above	0.61	0.49–0.76		0.50	0.39–0.63
Level of Completed Education, Direct Effects Analysis			sex, age, religion, household income, occupation, migrant status, roof, malaria history, malaria history in the family, IRS past 6 months, mosquito net, window net, repellent coil, repellent liquid, repellent skin cream		
None or below 10th grade	1	-ref-			
10th grade	0.43	0.34–0.54		0.45	0.30–0.69
Pre-university college or graduate and above	0.61	0.49–0.76		0.60	0.37–0.97
Migrant Status			sex, age, religion and education		
No	1	-ref-		1	-ref-
Yes	6.53	5.42–7.86		5.07	4.09–6.28
Migrant Status, Direct Effects Analysis			sex, age, religion, household income, occupation, education, roof, no. of people per room, malaria history, malaria history in the family, IRS past 6 months, mosquito net, window net, repellent coil, repellent liquid, repellent skin cream		
No	1	-ref-		1	-ref-
Yes	6.53	5.42–7.86		2.43	1.60–3.67
Roof type			education, occupation, migrant, household income		
Poor quality	1	-ref-		1	-ref-
Tiles	0.34	0.26–0.44		0.71	0.53–0.95
Cement	1.05	0.82–1.34		1.99	1.52–2.60
Indoors Residual Spraying in the Past 6 Months			education, being a migrant, roof type		
No	1	-ref-		1	-ref-
Yes	0.02	0.01–0.04		0.02	0.02–0.04

## Data Availability

The data presented in this study are available on request from the corresponding author.

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
