# Peer review of "The Effect of Socioeconomic Factors and Indoor Residual Spraying on Malaria in Mangaluru, India: A Case-Control Study"

_ijerph, 2021, doi:10.3390/ijerph182211853_

Round 1
Reviewer 1 Report
I want to thank the authors for submitted this interesting manuscript on the results of their case control study on the effects of socioeconomic factors on urban malaria. Here are a few comments that I hope can help improve the manuscript
Title - I think that the title should be changed to reflect the major finding from the study
Methods
- No information is provided is provided on the sample size
- It is not clear how IRS relates to roof type. IRS is typically provided by the malaria control program and not influenced by the household level decisions or housing characteristics? It is especially interested that very few patients reported IRS exposure, since educational attainment is low, could this just be that they didn't understand the question?
-Since the study estimates the direct effects of migration, why is that of educational attainment not estimated? This would be good to compare.
Author Response
Response to Reviewer 1
Comments and suggestions
Title - I think that the title should be changed to reflect the major finding from the study
Authors’ reply: We thank the reviewer for raising this point, but respectfully disagree. Since we have multiple findings that are influenced by various factors, we think a more descriptive title is most suitable.
Methods
No information is provided on the sample size.
Authors’ reply: The sample size is given in the results section, lines 183-184. In addition to that, we followed the reviewer’s suggestion and added the sample size and the effect size it allows to detect with sufficient power to the methods section in lines 109-113:
“Regarding the sample size, recruiting 800 malaria patients and 2000 controls allows for detecting associations of rare exposures (10% prevalence) at a strength as small as odds ratio (OR) 1.4, considering usual assumptions (80% power, 95% confidence interval). For exposures at 50% prevalence, this sample size facilities sufficient power to detect an OR as small as 1.2.”
It is not clear how IRS relates to roof type. IRS is typically provided by the malaria control program and not influenced by the household level decisions or housing characteristics?
Author’s reply: We appreciate the reviewer’s note on this and it is true that IRS is not decided on a household level. Some study participants reported to live in non-permanent residencies. Their non-registered living complexes might not be monitored by malaria control programs and IRS might not be applicable in those circumstances. Therefore, we assumed a possible influence between housing characteristics (e.g., roof type) and IRS. We changed the rationale in Table A1 accordingly. We would like to point out that in general in causal DAGs it is a stronger assumption to leave out an arrow than to add one in the correct direction, which is the rationale for being rather liberal with adding possible dependencies.
It is especially interested that very few patients reported IRS exposure, since educational attainment is low, could this just be that they didn't understand the question?
Authors’ reply: We consider it unlikely that the IRS question was misunderstood, because the questionnaires were taken by trained social workers.
Since the study estimates the direct effects of migration, why is that of educational attainment not estimated? This would be good to compare.
Authors’ reply: We welcome the reviewer’s suggestion and added the analysis of a direct effect of education to the manuscript. (Table 2, and Lines 149, 152, 227-229, 249)
Reviewer 2 Report
The manuscript is well written, presents enough data, the directed acyclic graph analysis made an interesting approach for summarize qualitative variables. After some major revision it could be suitable for publication.
I only some concerns. The first one is regarding the migrant status discussion. Authors proposed that migration can promote malaria itself, this by imported infections or increased (genetic) susceptibility. From an epidemiological point of view is just SUSCEPTIBLE INDIVIDUALS (neither imported nor increased susceptibility), migrants from free or low malaria areas moves to endemic urban areas (with current high malaria transition) with infrastructure and service deficiencies. So, a relevant question for further surveillance is “Had household member or close neighbors with malaria at any time BEFORE MIGRATION”. Authors could discuss this point.
Also the question “Had household member with malaria in the last 12 months” could be “Had household member with malaria AT ANY TIME”. I suppose that the original question is related with the IRS. However, do authors know the operation of the malaria programme in Mangaluru? This is relevant since the programme may have different strategies for applying the IRS. The first one is when a malaria case is confirmed, health-care workers spray (indoors) the case and surrounding houses in several rounds (with 6 months duration). The second is programmed spraying in risk areas (but they are actually the areas with confirmed cases with presence of mosquitoes). I do not understand if persons in the control group, that reported IRS, live in sprayed houses or near a spraying house, so “Indoor residual spraying in your house or near houses in the last 6 months” could be a more suitable question. Is probable the uninfected individuals live in areas with high spraying rate, affecting the probability to get infected. Meanwhile infected persons live in (recent created) areas where spraying is not that common.
It is possible to have a map with the risk areas and the proportion control and infected persons living in there?
Please, in table 1 include standard deviation for age, household income, number of persons in household, rooms in household, and persons per room in household. I am aware that data do not have normal distribution, however max-min does not gave a full comprehension of data variability. I might suppose that younger and active migrants could have more mobility in urban areas, so the probability to get infected increase.
Author Response
We thank the reviewers for their time and consideration in reviewing our manuscript. We carefully considered the comments and suggestions, and believe that the manuscript improved because of that.
Response to Reviewer 2
Comments and suggestions
Authors proposed that migration can promote malaria itself, this by imported infections or increased (genetic) susceptibility. From an epidemiological point of view is just SUSCEPTIBLE INDIVIDUALS (neither imported nor increased susceptibility), migrants from free or low malaria areas moves to endemic urban areas (with current high malaria transition) with infrastructure and service deficiencies. So, a relevant question for further surveillance is “Had household member or close neighbors with malaria at any time BEFORE MIGRATION”. Authors could discuss this point.
Authors’ reply: Thank you for this suggestion. Depending on the context, migration could cause an influx of susceptible individuals that are now newly exposed in a high transmission zone. However, in our study, this is likely not the case. The vast majority of migrants came from higher malaria transmission areas (northern and northeastern Indian states: West Bengal, Jharkhand, Uttar Pradesh, Bihar, Odisha, Assam), many of those reported to have had malaria before. This has been previously described in Gai et al, 2018, Malar J, https://doi.org/10.1186/s12936-018-2462-7. We now addressed this in the discussion, lines 258-262:
“Another possible explanation of migration status being a risk factor for malaria, is the potential influx of malaria susceptible migrants from low malaria transmission areas into a high malaria transmission area. However, in our study setting, this is likely not the case, because many migrants originated from areas where malaria prevalence exceeds that of Mangaluru [17].”
Also the question “Had household member with malaria in the last 12 months” could be “Had household member with malaria AT ANY TIME”.
Authors’ reply: We thank the reviewer for pointing this out, as we now realized we did not formulate this variable correctly in Table 1. This variable indeed describes whether participants had a household member that had malaria at any time. We corrected this accordingly. In the text, we do not mention any time limit on malaria history.
I suppose that the original question is related with the IRS. However, do authors know the operation of the malaria programme in Mangaluru? This is relevant since the programme may have different strategies for applying the IRS. The first one is when a malaria case is confirmed, health-care workers spray (indoors) the case and surrounding houses in several rounds (with 6 months duration). The second is programmed spraying in risk areas (but they are actually the areas with confirmed cases with presence of mosquitoes). I do not understand if persons in the control group, that reported IRS, live in sprayed houses or near a spraying house, so “Indoor residual spraying in your house or near houses in the last 6 months” could be a more suitable question. Is probable the uninfected individuals live in areas with high spraying rate, affecting the probability to get infected. Meanwhile infected persons live in (recent created) areas where spraying is not that common.
Authors’ reply: We appreciate the reviewer’s insights on this topic. Mangaluru’s vector control strategy is the routinely spraying of considered risk areas, including so-called focused spraying (i.e., IRS in households of a reported case). Our questionnaire did not include a question on neighborhood IRS, which is indeed a pity, since this could have affected the participants’ exposure. On the other hand, collecting such data by individual questionnaire could be incorrect. In retrospect, it would have been best to collect spatial data from the malaria control program regarding of IRS coverage. We added this now in the discussion, lines 306-309:
“A deficit in our data is the lack of IRS coverage by neighbourhood, which could have provided a better insight in the effect of IRS, since such vector control can have an effect beyond individual households.”
It is possible to have a map with the risk areas and the proportion control and infected persons living in there.
Authors’ reply: We thank the reviewer for this suggestion, and now added a figure displaying the malaria prevalence among controls, which illustrates the risk areas in Mangaluru. We refer to this figure in the results section, lines 192-193:
“…, the uneven distribution of which is illustrated in Figure 1.”
As for the recruited malaria patients, ward distribution was unequal too (almost 30% were recruited from 5 wards). The distribution of patients did roughly overlap by geographical location with the prevalence distribution in Figure 1, but not on ward level. We suggest to not include a map of patient recruitment by ward, because it might give an incomplete picture of the malaria cases distribution in Mangaluru (we recruited patients at the public hospital only).
Please, in table 1 include standard deviation for age, household income, number of persons in household, rooms in household, and persons per room in household. I am aware that data do not have normal distribution, however max-min does not gave a full comprehension of data variability.
Authors’ reply: We thank the reviewer for suggestion to add the standard deviations, and added the means and SD for numerical variables in Table 1 now.
I might suppose that younger and active migrants could have more mobility in urban areas, so the probability to get infected increase.
Authors’ reply: In a univariate analysis, malaria patients were younger than controls. In the multivariate analysis, we adjusted for age when investigating migrant status and our results imply that being a migrant affects malaria independently from age.
Round 2
Reviewer 2 Report
This is a really interesting paper, with a large data set and relevant epdiemiological survey. It should be accepted for publication at it stands.